# EMG Based Analysis of Gait Symmetry in Healthy Children

**DOI:** 10.3390/s21175983

**Published:** 2021-09-06

**Authors:** Kristina Daunoraviciene, Jurgita Ziziene, Jolanta Pauk, Giedre Juskeniene, Juozas Raistenskis

**Affiliations:** 1Department of Biomechanical Engineering, Vilnius Gediminas Technical University, 03224 Vilnius, Lithuania; kristina.daunoraviciene@vilniustech.lt (K.D.); jurgita.ziziene@vilniustech.lt (J.Z.); 2Biomedical Engineering Institute, Bialystok University of Technology, 15-351 Bialystok, Poland; 3Department of Rehabilitation, Physical and Sports Medicine, Health Science Institute, Faculty of Medicine, Vilnius University, 03101 Vilnius, Lithuania; giedre.juskeniene@santa.lt (G.J.); juozas.raistenskis@santa.lt (J.R.)

**Keywords:** gait symmetry, gait in children, surface EMG, muscular activation

## Abstract

The purpose of this study was to examine the changes in muscular activity between the left and right lower legs during gait in healthy children throughout temporal parameters of EMG and symmetry index (*SI*). A total of 17 healthy children (age: 8.06 ± 1.92 years) participated in this study. Five muscles on both legs were examined via the Vicon 8-camera motion analysis system synchronized with a Trigno EMG Wireless system and a Bertec force plate; onset–offset intervals were analyzed. The highest occurrence frequency of the primary activation modality was found in the stance phase. In the swing phase, onset–offset showed only a few meaningful signs of side asymmetry. The knee flexors demonstrated significant differences between the sides (*p* < 0.05) in terms of onset–offset intervals: biceps femoris in stance, single support, and pre-swing phases, with *SI* values = −6.45%, −14.29%, and −17.14%, respectively; semitendinosus in single support phase, with *SI* = −12.90%; lateral gastrocnemius in swing phase, with *SI* = −13.33%; and medial gastrocnemius in stance and single support phases, with *SI* = −13.33% and −23.53%, respectively. The study outcomes supply information about intra-subject variability, which is very important in follow-up examinations and comparison with other target groups of children.

## 1. Introduction

Walking is one of the most fundamental human activities and is strongly connected with health, physical condition and mental state. Gait analysis and assessment is a systematic, clinical study of how a person moves through the swing and stance phases of their gait; it is performed to assess for disturbances, associate them with other symptoms, and select the best treatment modality or technical aids. With the help of today’s motion analysis systems, it is possible to obtain extremely accurate gait kinematic data, calculate kinetic parameters, perform different levels of analytical calculations and predict the effectiveness of treatment. Surface electromyography (EMG) comprises a significant part of gait analysis. EMG supports clinicians via objective assessment of muscular function during walking. Muscle contraction and co-contractions are predominantly relevant in analyzing children’s pathologies [1]. EMG is regularly used to assess the activation patterns of various muscles of the lower limbs during both normal and pathological gait [2,3,4,5,6]. However, accurately assessing the parameters of normal gait and the behavior of the musculoskeletal system requires more than advanced technology and the latest tools; it is important to know exactly how to assess abnormalities. Typically, changes in gait or movement are defined by deviations from the norm, i.e., the typical motion standard [7]. However, how does one precisely determine what a normal gait is? Movement rates vary due to a number of factors that may directly affect function of the musculoskeletal system. Normal gait requires strength, balance, sensation, and coordination, while muscle activity also plays an important role. Slight variations exist in the normal gait and EMG patterns of individuals, especially children [1,2]. Analyses showing the effect of gender on EMG signals in adults [8] and school-age children [9] have been reported in the literature.

Growth factor is also known to influence gait stability and coordination [10,11]. In addition to growth, a maturation process resulting in gait stabilization has been identified between 3.5 and 4 years of age [12,13,14]. Gait speed is another gait characteristic that has been studied to help clarify gait biomechanics and regulation of the musculoskeletal system in situations where gait slows significantly. Extreme reductions in walking speed cause changes in locomotion, which may eventually result in modifications of the muscle activity patterns underlying gait [15,16]. Therefore, when assessing pathological gait, it is a challenge to identify which changes are already very serious and which are mild; such decisions are subjectively made by visual assessment of gait symmetry and stability. Nevertheless, most studies investigate gait patterns relating to specific pathologies or injuries. There is evidence of an increasing effect of gender and age on muscle recruitment [12,14] and walking speed on muscular activity patterns [15,16,17]. However, there is currently a lack of normative data regarding EMG patterns of activation; in most cases, only a few references concerning data of adults and children [1,2,12,18] are taken into account by clinicians. Previous analyses have quantified the repeatability of the EMG waveform in adult subjects; however, EMG variability in the pediatric population may significantly differ [8]. Rosengren et al. [19] have shown that children with conditions such as coordination disorder exhibit significant asymmetry when walking on a treadmill at around 0.85 m/s. Moreover, recent studies have identified gait patterns connected to asymmetric behavior of lower leg muscle recruitment during walking in children with cerebral palsy [20]. In general, most studies of normative gait in children have not reported measures of gait symmetry supported by surface EMG. There, the present study focused on gait symmetry characterizations in healthy children in order to establish EMG timing instances during walking at a self-selected speed. 

## 2. Materials and Methods

### 2.1. Subjects and Procedure

The study involved 17 healthy kids aged 4–11 years. Seventeen eligible typical children, aged 8.1 ± 1.9 and BMI 16.4 ± 2.2, were recruited. Demographic and descriptive data for the children are shown in Table 1.

The experimental protocol was approved by the regional ethical review board (No. 2020/9-1256-738). Parental consent and child assent were obtained prior to participation in the study. Criteria for inclusion in the study were: no injuries or illnesses of the musculoskeletal system, orthopedic surgeries, or gait abnormalities and a BMI ≤ 22.9. The following anthropometric data were measured for each subject: lengths of the thighs, lower legs, feet, and entire legs (from the anterior superior iliac spine to the malleolus medialis) and the widths of the pelvic, knee, and ankle joints. These data were used for the static calibration of the Plug-in-Gait model. Each subject was given an ID number, and a demographic and anthropometric data questionnaire was completed. Thirty-nine Vicon reflective markers were then fixed on the subject’s body comprising the full body Plug-in-Gait marker set. Since five muscles per leg were selected for examination, biceps femoris (BF), rectus femoris (RF), semitendinosus (SE), lateral gastrocnemius (LG), and medial gastrocnemius (MG), 10 EMG sensors were attached to the skin with special disposable stickers (Figure 1). 

The start and end of the 7 m trail were clearly marked by sticky tape lines. The child was asked to walk barefoot, at a comfortable speed, from one line to the other; this was considered a single measurement. Measurements were repeated at least 10 times for each subject. A total of 171 trials were further processed, equating to an average of 10.06 ± 3.44 trials per child. At least two consecutive strides were analyzed per trial, resulting in an average of 20.11 ± 6.87 strides per child; the total number of all trials’ strides was 316.4 ± 6.47 for the right leg and 336.4 ± 10.14 for the left leg. 

### 2.2. Signal Acquisition and Processing

Motion data were collected by eight cameras using the Vicon motion capture system (Oxford Metrics Group, Oxford, UK) at 100 Hz sampling rate. Ground reaction forces were obtained by a force plate (Bertec, Columbus, OH, USA, load capacity 5000 N) at 1000 Hz sampling. Various phases of the gait cycle, consisting of stance, swing, pre-swing, double support, and single support, were defined for the right and left legs. EMGs were recorded using the wireless Trigno EMG (Delsys, MA, USA) system at a sampling rate of 2000 Hz. Small wireless EMG sensors contain four silver bar electrodes and integrated amplifier. Skin was prepared using abrasive gel and cleaned using isopropyl alcohol to lower skin impedance. No gel was required, allowing fast, simple and hassle-free electrode placement. The reusable sensors were directly attached to the skin using double sided adhesive tape. Marker tracking, ground reaction forces, and EMG registration were all synchronized. The EMG data collected were pre-processed in the following order: (1) filtered with a high-frequency 4th order zero-phase Butterworth filter with a 50 Hz cut-off frequency; (2) filtered with a zero-lag low-frequency 2nd order Butterworth filter with a 6 Hz cut-off frequency; (3) cropped from heel contact to heel contact to strides; (4) transformed using the Teager-Kaiser Energy Operator (TKEO) [21]; (5) subjected to full-wave rectification; and (6) smoothed. Subsequently, the EMG curves were normalized to full gait in a random two-stride sequence for each test and the time was normalized to 100%. Various methods have been used [22,23] to perform threshold set-up; however, the best fit for our data was selected using the optimal threshold for detection of muscle activation on/off values, which was found to be 35% RMS [24]. The results of the EMG signal processing are presented in Figure 2.

The actual number of muscle activations, also known as activation modalities, in a single gait cycle was identified and averaged. We then further analyzed a single maximal amplitude activation modality. The activation amplitude was evaluated in percent, with values between 61% and 90% being considered as high-level activation [25] and taken as primary activation. The average occurrence frequencies of muscular recruitment were related to all of the detected primary activation modalities. The occurrence frequency of a primary activation was quantified by the number of strides in which the muscle was recruited with its highest peak modality in the phase in respect to the total number of strides analyzed [20]:(1)$$f_{i,s}=\frac{n_{ph}}{n}\times 100\;\%,$$
where: *f* = occurrence frequency, *i* = muscle, *s* = leg side, *n_ph_* = number of strides with the primary modality in particular phase (stance, swing, double support, single support, and pre-swing), and *n* = total stride number for the muscle. 

Eventually, the onset (ON)/offset (OFF) time instants of primary activation modality were averaged over the population of subjects, while the ON–OFF interval was defined as the activity duration as percentage of each gait phase for each observed muscle. Symmetry Index (*SI*) was calculated as an interlimb differences measure. *SI* for activation duration (ON–OFF interval) was calculated through the following equation:(2)$$SI=\frac{\left(\mathrm{DR}\;-\;\mathrm{DL}\right)}{0.5\times \left(\mathrm{DR}\;+\;\mathrm{DL}\right)}\times 100\%,$$
where: *SI* = Symmetry Index, DR = mean value of ON–OFF interval for the right side, and DL = mean value of ON–OFF interval for the left side. *SI* corresponds to the percentage of asymmetry observed for one side in relation to the other [26]; an *SI* = 0 indicates the existence of perfect symmetry, while positive and negative values assume that the asymmetry applies to the right and left sides, respectively. 

### 2.3. Statistical Analysis

The Lilliefors normality test (*p* < 0.05) was used to test data normality. The results for the right and left lower limbs were compared. Normally distributed data were compared utilizing the parametric statistical method, i.e., two sample paired *t*-test (*p* < 0.05); data that were not normally distributed (*p* < 0.05) were compared by employing a non-parametric statistical method, i.e., the Wilcoxon signed rank test (*p* < 0.05). With regard to the relationship between quantitative variables the normally distributed data are represented as mean ± SD, while the non-normally distributed data are represented by median (MAD, IQR). All analyses were performed using Matlab R2019b software (MathWorks Inc, Apple Hill Drive Natick, MA 01760, USA).

## 3. Results

### 3.1. Primary Activation Modality and Time Parameters

The first thing we noticed is that, in each subject, the muscles showed different numbers of activations during the same walk. Detailed analysis of the actual number of muscular activations showed that there may be anywhere from one to a maximum of five peaks of muscle activity per gait cycle, and that the duration and amplitude of these activations vary widely. Therefore, the averaged actual number of muscular bursts was calculated for each muscle, as well as for the right and left sides (Table 2). The largest difference between the right and left sides was found in the RF muscle, while the smallest difference was in the LG.

The calculated durations of averaged muscle activity in the gait cycle showed significant differences (*p* < 0.05) between right and left legs for the BF and MG muscles. The duration of primary activation per one full cycle, represented in medians (MAD), is shown in Table 3. Bold indicates significant difference (*p* < 0.05) right vs. left leg.

For greater accuracy, the entire cycle was divided into phases with limits as indicated in Table 4. The heel strike and toe off were determined and all spatiotemporal parameters were calculated. No significant differences were observed in the phases between right and left sides (*p* > 0.05). The ranges presented in Table 4 meet the norms for healthy individual [27]. 

Once the main activations were identified in each cycle, their time parameters were analyzed. To find out in which gait phase the main activity most often occurred, we defined how all the main activations found are distributed in the gait cycle (Figure 3). From the analysis of the two main phases of the cycle, i.e., stance and swing, the muscles are most active in the stance phase, in which 46.01% up to a maximum of 58.06% of all activations occur. However, no significant difference was observed between the phases; thus, it is clear that all major activations were distributed approximately equally between the two phases. The percentage differences between corresponding muscles on each side varied from a minimum of 0.25% to a maximum of 8.02%. The highest difference between sides was found in the pre-swing phase of BF. Meanwhile, RF demonstrated a high difference of 6.96% between the right and left sides in the stance and swing phases. The most stable manifestation of activity bursts between right and left sides was found for the SE, which showed a maximum difference of 4.04% during the pre-swing phase.

Onset and offset instances in each phase, for both the right and left sides, are presented in Table 5. Activation appears to begin earlier in the gait phases in the left leg of all muscles tested. Significant onset and offset differences between the right and left legs were obtained for all muscles except RF in the stance phase. In contrast, the swing phase was more stable, with only LG demonstrating a meaningful difference in ON and OFF instances.

Onsets and offsets of the activations, with durations, are shown for the right vs. left sides in one gait cycle in Figure 4. It is clear that muscle activity lasts for the several phases of gait. The figure represents a distribution of muscular activity pattern in terms of temporal parameters on the right and left sides. Time instances are as medians without variation for clearness. IQR and MAD are provided in Table 5 and Table 6. 

### 3.2. Symmetry Indices

The descriptive values of activation duration and *SI* for each muscle are presented in Table 6.

The results of the *SI* values show that gait symmetry is achieved by balancing between the sides (Table 6). BF demonstrated significant differences in the duration of the stance, single support, and pre-swing phases between the right and left legs (*p* < 0.05), with *SI* values equal to −6.45%, −14.29%, and −17.14%, respectively. SE demonstrated a significant difference in the interval results for the single support phase (*p* < 0.05), with *SI* = −12.90%. LG showed significant differences in duration in the swing phase (*p* < 0.05), with *SI* = −13.33%, while MG had meaningful differences in the stance and single support phases (*p* < 0.01), with *SI* = −13.33% and −23.53%, respectively. Ideal symmetry between the right and left sides, i.e., *SI* = 0, was achieved for SE in swing phase and for LG in double support phase. Overall, a slightly larger asymmetry was indicated towards the left side. Accordingly, if the BF activity interval is longer on the left side in the stance phase, then the RF counterparts on the right are related to the interaction of this muscle pair, as are RF vs. SE in the stance phase and the synergists LG and MG in the swing phase (Figure 5).

## 4. Discussion

It is common to assume that the gait of a healthy person is always symmetrical, and this assumption is usually made without any discussion in the literature. Any deviation from a symmetrical gait is considered a disorder and is attributed to a disfunction of the appropriate nature. Symmetry requires equilibrium and good stability of the core muscles and the joints, particularly the hip, knee, and ankle. To our knowledge, our study is novel in that EMG-based gait symmetry has not previously been studied in healthy children. Nardo et al. [28] presented results from an asymmetry study in children with mild hemiplegic cerebral palsy. Specific gait patterns were found, and asymmetry behavior was explained. Differences between the right and left legs have previously been observed [29], recognizing that right or left dominant preferences may cause small asymmetries. However, these deviations must typically reduce total system performance that the multi-joint segment must therefore compensate for. Symmetrical EMG patterns were observed for muscles during walking without objective documentation. However, in these studies, EMG data were either collected from only the dominant limb or a single lower limb, or the side was not specified [29]. 

### 4.1. Muscular Activation

The analysis of a high number of strides for each participant allowed us to obtain the bigger picture of muscular activation occurrence between right and left lower limbs during gait, demonstrating that a subject uses a specific muscle with different activation size even in the same walk. Our tasks were to find out how these activation intervals were arranged between sides to maintain stability and synchrony. All children in the experiment appeared to walk stably, confidently, and symmetrically from the naked eye. Therefore, we expected to find approximately equal characteristics of each muscle activity on both sides. We found that each subject’s muscle may generate a different number of activity intervals in the gait cycle. We have found that the number of activations may increase from 1 to 5, thus confirming previous observations [1,9,18]. During walking, the number of muscle activations within a cycle is cycle dependent: it may vary from stride to stride. Large variations of obtained results between subjects and between gait cycles made the results difficult to compare. The literature presents muscle activation intervals throughout the gait cycle, resulting in large variations in results [30]. Looking at our data, we realized that the onset–offset interval could be anywhere in the gait cycle. Therefore, we divided the gait cycle into phases [9,18,25,31] and determined the ranges of muscle activity within them. This completely circumvented the previously mentioned problem and the variations were diminished, which allowed us to more accurately assess the differences and compare results. The stance phase, demonstrated the highest percentage of muscular activations taking place within it. However, we observed that activation was distributed evenly between the stance and the swing phases depending on the time span of each phase. This indicates a stable gait. The difference in the occurrence frequency of primary activations between the left and right was obtained uniformly for the RF, SE, and LG muscles in the stance and swing phases, indicating synchronization of muscle function between these major gait phases. If the right RF activated more in the stance phase, so too did the left in swing phase. 

### 4.2. Temporal Parameters and Symmetry Indices

Moreover, the results of onset and offset (Table 4) instances demonstrated that the muscle activity intervals in healthy children significantly differ (*p* < 0.05) between right and left legs. It was observed that the activity of all studied muscles, with the exception of RF, activated earlier in the left leg (*p* < 0.05) in the stance phase, while the gap between the left and right legs decreased in the swing phase. Analyzing the full on and off interval (Table 5), we see that the differences between the sides clearly decreases, with less significant differences demonstrated. In this way, a balance is achieved between the left and the right with only a few signs of asymmetry. BF demonstrated meaningful ON–OFF intervals between the right and left legs in stance, single support, and pre-swing phases (*p* < 0.05), with *SI* values of −6.45%, −14.29%, and −17.14%, respectively. SE showed significant interval results in single support (*p* < 0.05) with *SI* = −12.90%. LG showed significant duration asymmetry in the swing phase (*p* < 0.05), with *SI* = −13.33%, while MG demonstrated meaningful differences in stance and single support phases (*p* < 0.01), with *SI* = −13.33% and −23.53%, respectively. The ideal symmetry between sides, *SI* = 0, was achieved for SE in swing and for LG in double support. Summarizing our results, we can say that there is no ideal symmetry in children’s gait; however, balance is achieved by an arrangement between the sides. Our results showed that asymmetry more towards the left side with muscles recruited earlier; however, in the sequential phases, more symmetry was achieved due to muscle interactions. It is difficult to notice these changes visually, proving that the asymmetry obtained by us is of little significance for children’s gait stability. We believe that the left-side dominance in the results can be explained by compensation for the dominant side. However, we are only guessing because we did not identify the dominant side of the child in our study. We also notice a balance between muscle pairs, e.g., BF and RF, RF and SE, as well as LG with MG. Therefore, we intend to investigate the interaction between these muscles, since muscle contraction and co-contractions seem to be predominantly relevant.

## 5. Conclusions

The analysis of these temporal EMG parameters is helpful for characterization of children’s gait and quantifies how often children exploit a specific gait pattern and how it differs between legs. Our study results also supply information about intra-subject variability, which is very important in follow-up examinations and comparison with other target groups of children.

## Figures and Tables

**Figure 1 sensors-21-05983-f001:**
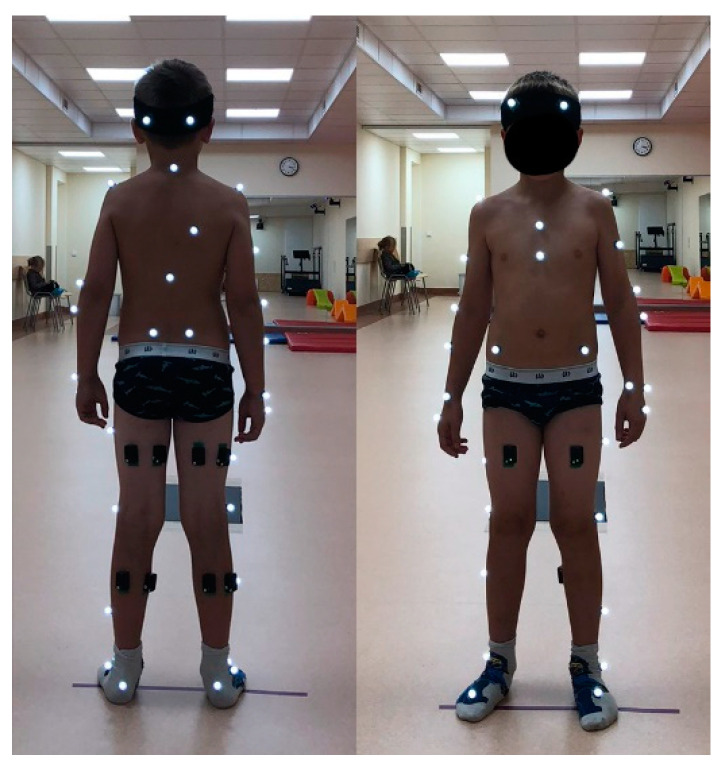
Children gait measurement setup with 39 Vicon reflective markers (indicated by white dots) and 10 EMG sensors attached by black disposable stickers at both lower legs. Start and end of the 7 m trail are marked by sticky tape lines. Children walked barefoot during measurements. The children wore socks and wore light clothing outside of the experiment.

**Figure 2 sensors-21-05983-f002:**
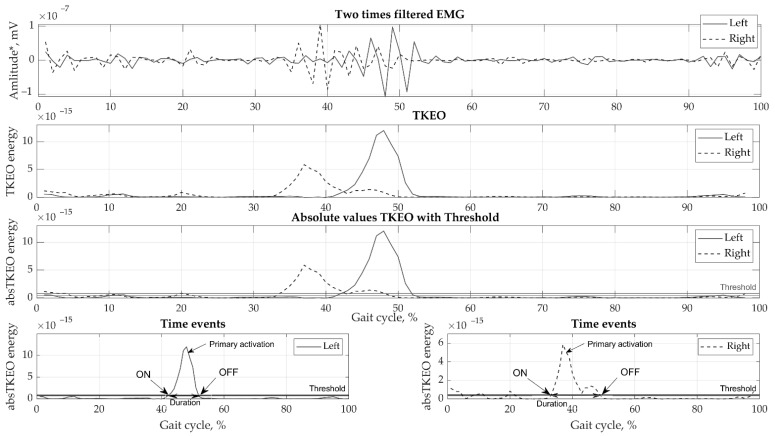
Example of EMG processing steps showing a single biceps femoris sample.

**Figure 3 sensors-21-05983-f003:**
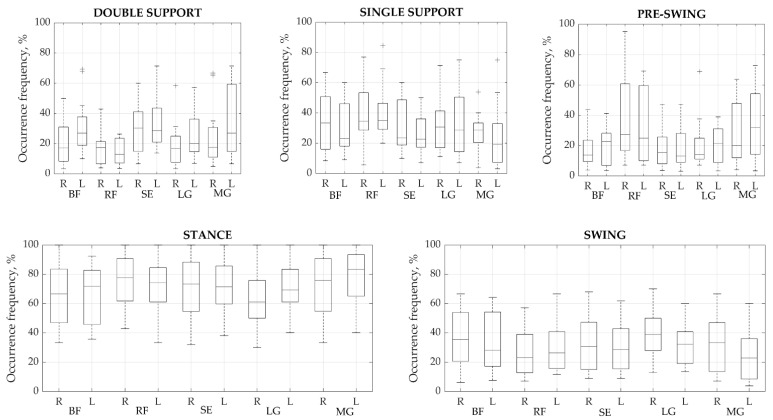
Occurrence frequency of a primary activation modality for each muscle group during each phase (%). Values presented as means ± SD. BF—biceps femoris, RF—rectus femoris, SE—semitendinosus, LG—lateral gastrocnemius, and MG—medial gastrocnemius. R—right, L—left side.

**Figure 4 sensors-21-05983-f004:**
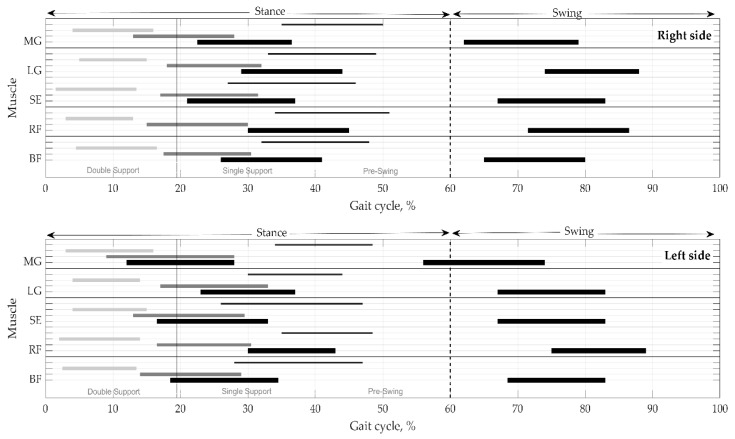
Distribution of muscular activation pattern on the right and left sides. Values presented as medians without mad. Thick black lines indicate the mean duration of activity in the main phases—stance and swing. Thin black, thick gray, and light gray lines reflect the mean values of activations in the subphases, respectively, in pre-swing, single support, and double support. BF—biceps femoris, RF—rectus femoris, SE—semitendinosus, LG—lateral gastrocnemius, and MG—medial gastrocnemius.

**Figure 5 sensors-21-05983-f005:**
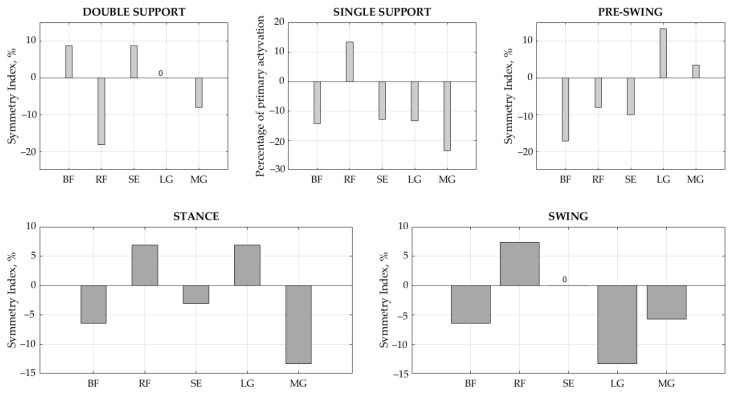
Symmetry index in each gait phase for each muscle. BF—biceps femoris, RF—rectus femoris, SE—semitendinosus, LG—lateral gastrocnemius, and MG—medial gastrocnemius.

**Table 1 sensors-21-05983-t001:** Demographic and descriptive data of the participants, *n* = 17.

	Age (years)	Height (cm)	BMI (kg/m^2^)	Pelvic Width (mm)	Knee Joint Width (mm)	Ankle Joint Width (mm)	Leg Length (mm)	Shoulder Offset (mm)
Male (*n* = 6)	9.0 ± 1.6	134.0 ± 10.9	15.9 ± 1.8	213.3 ± 31.4	86 ± 8.7	59.8 ± 4.6	725.0 ± 80.6	49.2 ± 16.6
Female (*n* = 11)	7.6 ± 2.0	129.9 ± 10.8	16.6 ± 2.4	205.0 ± 42.6	83.8 ± 9.6	56.3 ± 4.7	609.5 ± 72.2	39.8 ± 5.5
All	8.1 ± 1.9	131.4 ± 10.7	16.4 ± 2.2	207.9 ± 38.2	84.6 ± 9.1	57.5 ± 4.9	702.7 ± 74.7	62.4 ± 6.1

Data presented as mean ± SD.

**Table 2 sensors-21-05983-t002:** Number of activation bursts per cycle.

Muscle	Actual Number of Activations per Cycle		Difference (Activation)
	Right	Left	
BF	2.55	2.68	0.13
RF	3.29	2.94	0.35
SE	2.83	2.66	0.16
LG	2.67	2.58	0.09
MG	2.16	2.32	0.16

**Table 3 sensors-21-05983-t003:** Average (SD) activation duration of each muscle per full cycle for the right and left legs.

Muscle	Primary Activation Duration per Cycle, %	
	Right Leg	Left Leg
BF	**15 (5.10)**	**16 (5.79)**
RF	14 (5.14)	14 (5.14)
SE	16 (5.97)	16 (6.21)
LG	14 (4.66)	15 (5.09)
MG	**16 (6.06)**	**17 (5.62)**

Bold indicates significant difference (*p* < 0.05) right vs. left leg. BF—biceps femoris, RF—rectus femoris, SE—semitendinosus, LG—lateral gastrocnemius, and MG—medial gastrocnemius.

**Table 4 sensors-21-05983-t004:** Ranges of phases in one gait cycle.

Phases *	Right Leg	Left Leg
	Mean ± SD	Mean ± SD
Stance phase (double support + single support + pre-swing; %)	0–59.7 ± 4.9	0–59.3 ± 4.8
Double support (heelstrike + load response; %)	0–19.5 ± 9.0	0–19.5 ± 8.9
Single support (%)	19.5 ± 9.0–40.2 ± 5.1	19.5 ± 8.9–39.7 ± 5.5
Pre-swing (%)	40.2 ± 5.1–59.7 ± 4.9	39.7 ± 5.5–59.3 ± 4.8
Swing phase (stance phase–100; %)	59.7 ± 4.9–100 ± 4.9	59.3 ± 4.8–100 ± 4.8

* In Vicon’s software Plug-in-Gait model, heel strike and toe off for all steps of both feet were found from the force plate and heel and toe markers trajectories. According to the same heel strike and toe off (gait phases), the EMG data were cut out and analyzed further in this study.

**Table 5 sensors-21-05983-t005:** Time parameters for primary activation of each muscle in each gait phase.

Muscle	Time Inst.	Stance (%)		Swing (%)		Double Support (%)		Single Support (%)		Pre-Swing (%)	
		Right	Left	Right	Left	Right	Left	Right	Left	Right	Left
BF	ON	**26 (9.9)**	**18.5 (9.3)**	73 (10.4)	71 (11.9)	4.5 (2.6)	2.5 (2.0)	17.5 (7.8)	14 (6.5)	**32(6.9)**	**28 (6.5)**
	OFF	**43 (9.7)**	**38.5 (10.14)**	91 (9.2)	89 (10.1)	**16 (1.7)**	**13.5 (2.5)**	31.5 (5.6)	30 (5.3)	**48.5(4.5)**	**47 (4.2)**
RF	ON	30 (11.2)	30 (11.3)	62 (14.63)	69 (16.3)	2 (2.1)	2.5 (1.4)	15 (8.5)	15.50 (7.4)	34(6.3)	35 (5.7)
	OFF	47 (11.2)	46 (11.3)	79 (12.8)	86 (13.6)	14 (2.5)	15.5 (3.2)	34 (7.2)	26.50 (6.1)	52.5(4.4)	50 (4.4)
SE	ON	**21 (8.9)**	**16.5 (9.0)**	67 (11.5)	67 (11.2)	**1.5 (2.1)**	**4 (1.9)**	**17 (5.4)**	**13 (6.2)**	27(7.3)	26 (7.4)
	OFF	41 (9.5)	37 (10.5)	86 (9.8)	85 (8.7)	14.5 (2.4)	15.5 (2.2)	33 (5.2)	31.50 (4.9)	47(4.1)	47 (4.4)
LG	ON	**29 (9.9)**	**23 (10.4)**	**74 (10.4)**	**67 (11.6)**	5 (2.8)	4 (2.0)	18 (7.5)	17 (7.2)	**33(5.4)**	**30 (5.5)**
	OFF	**44 (9.8)**	**38 (9.9)**	**90 (9.5)**	**86 (9.8)**	15 (2.5)	15 (2.0)	32 (5.3)	34 (4.2)	**48(4.1)**	**46 (4.2)**
MG	ON	**22.5 (12.5)**	**12 (11.6)**	56(12.6)	56 (12.7)	4 (3.1)	3 (2.2)	**13 (5.3)**	**9 (5.8)**	35(6.0)	34 (5.6)
	OFF	**37 (11.7)**	**32 (10.8)**	80 (9.9)	77 (9.8)	17 (2.4)	16 (1.7)	30 (5.3)	30 (5.0)	49(4.3)	48 (3.7)

Values presented in median (MAD). Bold indicates a significant difference (*p* < 0.05) between the right and left sides. Biceps femoris (BF), rectus femoris (RF), semitendinosus (SE), lateral gastrocnemius (LG), medial gastrocnemius (MG), right leg (R), left leg (L). ON stands for the onset and OFF—for the offset of muscular activation.

**Table 6 sensors-21-05983-t006:** Descriptive values of activation duration and *SI* for each muscle.

Phase	Parameter	BF		RF		SE		LG		MG	
		Right	Left	Right	Left	Right	Left	Right	Left	Right	Left
Stance (%)	Duration (%)	15 (5.0)	16 (5.7)	14 (5.3)	13 (4.5)	16 (5.9)	16.50 (6.4)	15 (4.6)	14 (4.7)	14 (4.6)	16 (4.4)
	IQR 25%	11	12	11	11	12	13	11	11	12.50	10
	IQR 75%	19	22	20	19	22	23	18	19	20	19
	Wilcoxon	**0.048**		0.184		0.352		0.944		**0.001**	
	*SI* (%)	−6.45		14.29		−3.08		6.89		−13.33	
Swing (%)	Duration (%)	15 (5.2)	16 (5.9)	15 (5.4)	14.50 (6.1)	16 (6.0)	16 (6.0)	14 (4.7)	16 (5.5)	17 (6.9)	18 (6.7)
	IQR 25%	12	12	11	11	12	11	12	12	12	13
	IQR 75%	19.25	22	19.75	20	22	21.25	19	21.75	24.25	25
	Wilcoxon	0.105		0.771		0.269		**0.026**		0.343	
	*SI* (%)	−6.45		3.39		0		−13.33		−5.71	
Double support (%)	Duration (%)	12 (3.2)	11 (2.0)	10 (2.5)	12 (2.6)	12(1.9)	11(1.8)	10 (2.5)	10 (1.6)	12 (2.6)	13 (1.8)
	IQR 25%	7	9	9	10	11	10	8	10	10	12
	IQR 75%	14	13	14	14	14	12.50	13	12	13.75	15
	Wilcoxon	0.845		0.237		0.064		0.555		0.107	
	*SI*,	8.70		−18.18		8.70		0		−8.00	
Single support (%)	Duration, %	13 (4.2)	15 (4.5)	15 (4.6)	14 (3.1)	14.50 (4.2)	16.50 (3.8)	14 (4.0)	16 (4.6)	15 (4.7)	19 (4.3)
	IQR 25%	11	12	10.25	10	12	14	11	12	11	15
	IQR 75%	17	20	19	16	19	20	17.25	20	20	23
	Wilcoxon	**0.021**		0.425		**0.037**		0.056		**0.0003**	
	*SI*	−14.29		6.90		−12.90		−13.33		−23.53	
Pre-Swing (%)	Duration, %	16 (5.4)	19 (6.4)	17 (5.4)	13.50 (5.3)	19 (6.98)	21 (7.9)	16 (4.8)	14 (5.0)	15 (4.9)	14.50 (4.3)
	IQR 25%	12	14	12	11	14	15	12	12	10.75	11
	IQR 75%	21.50	25	21	20	27.50	31	19	21.25	19	19
	Wilcoxon	**0.031**		0.157		0.176		0.813		0.626	
	*SI*	−17.14		22.95		−10.00		13.33		3.39	

Values are presented as median (mad); bold indicates significant difference (*p* < 0.05) right vs. left. BF—biceps femoris, RF—rectus femoris, SE—semitendinosus, LG—lateral gastrocnemius, and MG—medial gastrocnemius.

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
