# Peer review of "EMG Based Analysis of Gait Symmetry in Healthy Children"

_sensors, 2021, doi:10.3390/s21175983_

Round 1

Reviewer 1 Report

Thank you for the interesting paper, comments can be found in the attached PDF.

Author Response

Dear Reviewer,

We are very grateful for your accurate observations and competent comments, which have greatly helped us to improve the quality of the publication. We tried to take every comment into account and step by step corrected our article. We hope that we have we responded efficiently and improved our article enough to make it suitable for scientific publication.

Sincerely,

Authors

Reviewer 2 Report

The work presents the analysis of symmetry based on muscular timing on a set of lower limb muscles in healthy children. The results showed some difference on both legs for the activation to characterize the gait in children. The authors suggest that the dominant side may cause compensations and produce asymmetry, but they recognize that this information was not collected in the experiments. In general, this study may be interesting to understand pathologies and differences in children’s groups. I left some concerns in order to improve the work.

Methods:

Trigno EMG equipment provides an EMG signal for each sensor. How was the configuration used to require two sensors for each muscle during the acquisition?

Results

The number of activation bursts per cycle may be affected by factors like sudden low amplitudes of the signal due to the stochastic nature of the EMG. It is clear the relevance of the activation intervals for the symmetry calculation. It was reported a number of activations from 1 to 5, but in practice, what does this number represents in terms of symmetry?

Figure 4.: “*-*” indicates double support phase, “o-o” single support phase, and “x-x” pre-swing phase. For me, it was unclear the meaning of the variability of these phases for each muscle, since they were demarcated accordantly with the ranges in Table 4.

Discussion

In general, the discussion seems to be a bit confused. I suggest dividing it by topics for easier reading.

What does the ideal symmetry on the SE and LG in the double support phase suggest?

Minor concerns:

Table 4. Should the single support and pre-swing phases include two SD, for each edge? Also, SD is missing in pre-swing.

Figure 4. Please, use "SE" instead "S" in the vertical axis.

The lines from 340 to 347 seem to repeat the same included in Results.

Author Response

(The authors gave the same response as above.)

Round 2

Reviewer 2 Report

Thanks to the authors for considering the suggested modifications. I have no other comments on the manuscript. Congratulations to the authors for this nice research.